# Nutritional and Organoleptic Characteristics of Sausage Based on an Autochthonous Ecuatorian Fish, Old Blue (*Andinoacara rivulatus*)

**DOI:** 10.3390/foods13091399

**Published:** 2024-05-02

**Authors:** Marlene Medina, Rosa M. García-Gimeno, Juan Alejandro Neira-Mosquera, Alexandra Barrera, Guiomar Denisse Posada-Izquierdo

**Affiliations:** 1Faculty of Livestock and Biological Science, State Technical University of Quevedo, La María Experimental Campus Km 7 ½ vía Quevedo-El Empalme, Quevedo 120406, Ecuador; mmedina@uteq.edu.ec (M.M.); neiramosquera@uteq.edu.ec (J.A.N.-M.);; 2Campus of International Agri-food Excellence CeiA3, UIC ENZOEM, Rabanales Campus, Darwin Building, University of Córdoba, 14071 Córdoba, Spain; bt1gagir@uco.es

**Keywords:** fish farms, sausage, native fish, nutritional and organoleptic characteristics, sustainable development goals

## Abstract

The consumption of fisheries and aquaculture products has been increasing in recent decades, and it is necessary to strike a balance between production and sustainability in aquaculture; this is essential homework to support the demand for human food. This study aimed to investigate the sensory and nutritional characteristics of a sausage made from the fillet of the native fish Old Blue (*Andinoacara rivulatus*) to stimulate the local economy. A multifactorial A*B*C design was used, with Factor A being rearing systems (wild and farmed), Factor B being types of protein (quinoa meal and soybean meal), and Factor C being lipids of animal (pork fat) and vegetable (sunflower oil) origin. Highly significant differences were observed in all formulas, according to Tukey (*p* < 0.05). The highest protein percentage was with soybean flour at 11.24%, while quinoa flour had 10.80% of the product. In sensory characteristics, the best attributes were for texture in the mouth with a hedonic scale from 0 to 5, with 4.2 firmness being mostly acceptable, odor at 4.5, the aromatic attribute with the highest and best value, and color was the clearest at 4.3 acceptability. The flavor was 4.3, mostly more pleasant according to the tasters. The yield of farmed fillets was 23.16% compared to wild 13.89%, and the balance of sausage yield was 393 g with a yield of 76.33% of the total weight. Microbiological analysis of the sausage with the native species showed the absence of pathogenic microorganisms. A commercial shelf life of 30 days was also established for the sausage, demonstrating an added value through this processing, allowing its exploitation in areas further away from the fishing sites and, thus, a greater possibility of social development in the area.

## 1. Introduction

The consumption of fisheries and aquaculture products has been increasing in recent decades, reaching 20.2 kg per capita in 2020, and sustainable aquaculture development is essential to support the demand for aquatic foods. Ecuador is among the major fishing and exporting countries [1]. It has a high biodiversity of freshwater wild fish where artisanal fishing techniques and gear predominate, although often without adequate management and control, which affects the reproduction of native species and their quality and, therefore, their survival. The native Old Blue (*Andinoacara rivulatus*) is an autochthonous species, produced and consumed mainly in the rural areas of Quevedo and its surroundings. It is also on the verge of extinction due to unregulated over-consumption. Implementing fish farms could avoid their extinction, preserve them in nature, and provide good-quality protein to rural populations.

Several studies have been conducted reflecting the influence of the farmed and wild-rearing systems on fish production performance [2,3,4] as well as the protein and lipid content of feed in aquaculture [5] but not on Old Blue *(A. rivulatus*) or using local products to contribute to the circular economy. 

Fish is one of the essential foods in human nutrition as it is an important source of proteins, minerals, vitamins, and fatty acids, and omegas 3, 6, and 9 have innumerable beneficial effects on health, such as the prevention of cardiovascular diseases [6,7,8]

The sausage format is widely accepted, and its adaptation to fish meat can promote the consumption of this essential and beneficial product in the diet. Sausages are often perceived as food of poor nutritional quality. However, better use of raw materials, such as fish, could be essential to human nutrition due to their high acceptance by some audiences, such as children [9].

Several studies can be found on fish-based sausages such as Talang Queenfish (*Scomberoides commersonnianuus*) [10], tilapia (*Oreochromis niloticus*) [11], Argentine anchovy (*Engraulis anchoita*) [12], African walking catfish (*Clarias gariepinus*) [13], and shortfin scad (*Decapterus macrosoma*) [14]. However, none focus on freshwater fish such as the Old Blue (*A. rivulatus*). 

A sustainable food production system promotes incorporating local ingredients such as quinoa and soya. Hence, there is a need to investigate the impact of its inclusion in a meat product as versatile as sausage. Previous research [15] showed that adding quinoa protein increases moisture, ash, protein, and pH without changing the sensory characteristics of the sausage and improving fat stability during storage. Several sausage studies have reported that soy proteins increase hardness and cohesiveness [16,17], which in this case, is a significant factor when it is elaborated on based on fish flesh, which is so soft.

In this study, *Andinoacara rivulatus* fillets were used as raw material for the elaboration of Frankfurt-type sausage as an innovative manufacturing product, taking into consideration the breeding systems of the native fish mentioned (wild and farmed), types of vegetable proteins (quinoa/soybean flour), and types of lipids (pork fat and sunflower oil), recognizing the organoleptic, sensory, and nutritional characteristics that this native species in the industrialization proposal. This proposal is expected to generate a socio-economic impact with the valorization of the raw material by constituting an alternative for the collection of innovative derivative products that contribute to the development of artisanal fishing, industry, and social actors in the regional fishing sector.

## 2. Materials and Methods

### 2.1. Experimental Design

The samples were taken from two rearing systems: Aquaculture is the farming of aquatic organisms, which can be carried out in artificial facilities or not, although the most remarkable thing is that all stages of development are controlled in optimal conditions; and an artisanal fishing system consists of those wild populations that inhabit rivers naturally, without the intervention of man for their production, allowing considerable variability of the wild species.

The wild samples were collected in artisanal fishing areas used by local people, who were the collectors and collaborators in this research. Artisanal fishing obtained wild specimens of old blues from rivers of the Baba (Buena Fe canton) and the “La Estrella” (Mocache canton) sector, and the farmed fish were situated in Puerto Bajaña (Mocache canton) and the D’Veritas (Mocache canton) sector, Puerto Bajaña, all of which are in the Los Rios province.

An equal number of fish were collected monthly from each rearing system, for a total of 72 fish during the winter period from October to December.

After being eviscerated, washed, and cut into pieces, samples of fish fillets were stored in airtight bags at −2 °C.

A multifactorial design (A*B*C) was applied, where each factor consists of two levels: Factor A = Rearing system (Wild/Fish farm), Factor B = Type of protein (Quinoa meal/Soybean meal), and Factor C = Type of lipid (Pig fat/Sunflower oil), obtaining a total of eight formula. Each treatment was replicated three times.

### 2.2. Formulation and Processing of Sausage

Description of the Old Bluefish dressing process will be the following: 

After receiving the fish, it was scaled and eviscerated, beginning with dissection from the anus along the abdominal line. The fish was then washed thoroughly and cut into pieces, separating the loin and belly. These were weighed for yield.

The formulation developed for the sausage production based on fillets (loin and belly) of the native Old Blue (*A. rivulatus*) species is described in Table 1. The mixture was transferred into a stuffer, and 28-gauge synthetic cellulose packaging was placed in the nozzle and fed under pressure, resulting in sausages weighing 80 g per 18 cm.

The sausages were placed in a pot of boiling water for 10 min, and the external temperature was controlled to not exceed 80 °C until the product’s internal temperature reached 72 °C. At the end of the blanching process, the sausage was immediately removed and placed in a container with ice water. The sausages were then refrigerated, maintaining a temperature of 3–5 °C for further analysis.

### 2.3. Nutritional, Organoleptic, and Microbiological Characteristics

Twenty-four hours after the sausages were prepared, moisture, protein, fat, and ash were determined according to the methods of the Official Association of Analytical Chemists (AOAC): moisture by oven drying (110 °C) to a constant weight [18], protein by Micro-Kjeldahl [19], using Tecator equipment (Kjeltec System, 1002 Distilling Unit, 2006 Digestor, Sweden), fat by Soxtec System HT 1043 (Hoganas, Sweden) [20], ash by muffle ashing [21], and finally the lipid profile of saturated, monounsaturated, polyunsaturated, and trans fats [22]. These analyses were performed in triplicate.

The Ecuadorian Technical Standard [23] was followed for pH measurement. For this purpose, 10 g of fish loin was weighed, diluted with 100 mL of distilled water using a mortar and pestle, mixed with water, shaken until a liquid was obtained, and used to measure the pH with the potentiometer.

In the sensory analysis, consumer acceptance of the product was determined based on the attributes of mouth texture, odor, color, and flavor using a 5-point hedonic scale, where 5 is an excellent characteristic value and 0 is extremely poor, with the following descriptors: mouth texture = (firm, soft, and viscous), odor = (rancid, aromatic, and sea-foody), color = (clear, opaque, and dark), and flavor = (pleasant, salty, and sour).

The sausages were heated with water to a temperature of 70 °C, cut into 2.5 cm portions, and identified with two-digit random numbers. Each evaluator was given a taste of one portion of each formulation; they were asked to eat a soda cracker and a sip of water between each. The evaluation was conducted in a ventilated, well-lit area, free of extraneous odors, with a panel of 12 trained evaluators, who were provided with an evaluation form.

The microbiological analyses (Ecuadorian Technical Standard [23]) determined molds, yeasts, total aerobes, total and fecal coliforms, Salmonella, Staphylococcus aureus [24], and histamine [25]. The results were compared with the complement of the criteria of the Ecuadorian Technical Standard [26].

### 2.4. Sausage Yield

During the production process of the Frankfurt-type sausage, the losses present in each process were considered; all the losses in each treatment were taken and weighed, registered after each emulsion as well as the final product, to determine the losses present in the product of the transformation. The following formula was used:$$\mathrm{Sausage}\;\mathrm{yield}=\frac{total\;weight\;of\;raw\;material}{weight\;of\;the\;sausage}\times 100\%$$

### 2.5. Statistical Analysis

The statistical program Statgraphics (Statgraphics Technologies, Inc., The Plains, Virginia, 2023) was used to determine the analysis of the variance of the study factors, and the interaction between formulas was determined using Tukey’s test (*p* < 0.05).

## 3. Results and Discussion

The sausage’s final weight yield was obtained based on the quotient of the total weight of raw material per treatment (393 g) over the weight of the sausage obtained (300 g), with a yield of 76.33%. Significant differences using Tukey’s test were observed between the rearing systems: the wild system was 62.4% and the farmed system was 81.7%.

The yield of the raw material used for sausage production varied significantly (*p* < 0.005) according to the rearing system: wild (43.23%) and farmed (37.05%). Reference [27] described highly significant differences (*p* < 0.001) between the yield of wild species (29.09%) and farmed species (31.94%), with the latter being higher.

Likewise, the best yield values obtained in the final product for the different formulations were “quinoa flour + pork fat” with 62.4% and “soya flour + sunflower oil” with 81.7%.

### 3.1. Nutritional Characteristics of Sausage

Table 2 and Table 3 show the results obtained for the immediate components, where we can conclude that Factor A (Rearing system) and Factor B (Type of protein) present highly significant differences. In contrast, Factor C (Type of lipid) did not show a significant difference. The values collected were like those described by [28] in fish sausage based on mackerel surimi, with values for moisture (54.47%), ash (1.90%), protein (20.12%), and fat (18.64%). Both results follow the Venezuelan Standard [29], which indicates 11% minimum protein values and 35% maximum fat. 

Furthermore, neither *Enterobacteriaceae* nor *Salmonella* was detected, ensuring the microbiological quality of the food. 

Ref. [14] described higher values for all immediate components, including % moisture content in their *Channa striata* sausages. On the other hand, Tilapia-based sausages [11] showed much higher values in all immediate principles except protein, which was half at a pH of 6.7. Meanwhile, [30] observed higher moisture and fat content, and protein and ash values were similar in pangasius (*Pangasius pangasius*)-based sausage.

The results obtained for ash content, which indicates the minerals present in the product, have been similar to other authors, such as [31], who describe values between 2.21 ± 0.10% and 3 ± 3.0%.

Ref. [32], who studied the Old Blue (*A. rivulatus*), the raw material of this study’s sausage, observed lower values of fat (4.15%) and ash (1.49%) and higher values of protein (21.14%) and moisture (75.77%).

Significant differences between rearing systems were observed for pH and moisture, with higher values in the wild-rearing system (Table 2). As for protein content, the fish farming system showed the highest value with statistical significance. There was no significant difference in the percentage of ash and fat. This contrasts with values observed by other authors, where the aquaculture system obtained significantly lower fat content in trout [33]. However, it describes much lower total fat values of 1.70 and 3.71%, respectively. [27] pH values were somewhat higher than those found here but in the raw material.

After the statistical analysis of the protein factor (Table 3), it was observed that in addition to the expected significant difference in protein content (*p* < 0.05) with mean values of 10.80 and 11.24 g/100 for quinoa and soybean flour, respectively, pH was also influenced with values of 6.06 and 5.96, respectively. Considering the moisture percentage in the sausage, soy flour obtained the highest value (Farmed 63.82%), unlike quinoa flour, which shows the lowest value (Wild = 63.59%). On the other hand, it was observed that soy flour showed higher values (*p* < 0.05) (Farmed = 2.49%) compared to quinoa flour, which showed the lowest value (Wild = 2.29%). Considering the data obtained for the fat percentage, no significant difference was found. 

Considering the result of the percentage of moisture present in the sausage, pork fat obtained the highest value (Wild = 65.54%), in contrast to sunflower oil, which showed the lowest value (Wild = 60.82%) (*p* < 0.05). In contrast to the protein determination, sunflower oil showed the highest value (Farmed = 11.38 g/100 g), while pork fat showed the lowest value (Wild = 10.65 g/100 g) (*p* < 0.05).

No significant difference was found in the data obtained for the percentage of ash and fat.

In the data obtained (Table 4) for the content of saturated fatty acids present, highly significant differences (*p* > 0.01) could be observed concerning the rearing system (wild/farmed). The nutritional characteristics of fatty acids beneficial to health are presented in the species of wild breeding with saturated fatty acids of 33.66%, and for fish farms, they presented at 29.13%. In monounsaturated fatty acids, it was observed that the wild species showed a higher content of 20.8% in contrast to the farmed species, which showed 16.16%. The polyunsaturated fatty acid content was 54.7% for the farmed species and 45.33% for the wild species. The latter coincides with those [34] described for *L. rohita*.

These observed values are higher and, in some cases, similar to those described by [35], who mention that monounsaturated acids are 32.9% of *Cyprinus carpio* fish and 27.2% of *Labeo rohita* fish and these stand out in palmitoleic and oleic, with polyunsaturated fatty acids being 11.4% of *C. carpio* fish and 22.2% of *L. rohita* fish and linoleic stands out. In our study, it was observed that in saturated fatty acids, Palmitic Acid (C16:0), monounsaturated fatty acids, Oleic Acid (C18:1n9cis), polyunsaturated fatty acids (Linoleic Acid (C18:2n6cis) (Omega 6), Docosadienoic Acid (C22: 2n6), and cis-11,14,17 eicosatrienoic acid (C20: 3n11)) were the components that presented the highest value. In Omega 6, the specimen of farmed origin showed 17.70%, while the wild species showed 11.60% in content. The predominance of palmitic acid coincides with that described by [34] for *L. rohita*. However, other studies in *L. rohita* [36] describe a higher percentage of lauric acid, followed by palmitic and stearic.

### 3.2. Microbiological Evaluation of the Sausage

The product was found to comply with the microbiological parameters established by the Ecuadorian Standard [23] during its shelf life due to the application of good manufacturing practices, good sterilization, and heat formula. Total and fecal coliforms, yeasts, molds, and pathogens were not detected (<1 CFU/g).

### 3.3. Organoleptic Evaluation of Sausage

The product’s sensory properties are crucial to achieving consumer acceptance of this new product. The sensory analysis was based on the texture in the mouth, smell, color, and taste of the sausage, with the factors “Wild + Quinoa flour + Sunflower oil” presenting the best sensory characteristics in all the parameters evaluated. In texture in the mouth (Figure 1) with a range of 0 to 5, it presented a firmness value of 4.2, making it acceptable, while for odor (Figure 2), it showed 4.5 in the aromatic characteristic, which was the highest and most acceptable value; for color (Figure 3), it was more evident with 4.3 acceptability, and finally, flavor (Figure 4) was 4.3, meaning it was mostly pleasant for the evaluators. These results agree with those described by [30] for Panga sausage, while the ratings described for tilapia sausage by [11] are much lower, with an overall acceptability of 3.2 on a 0–4 scale, making the Old Blue (*A. rivulatus*) sausage more palatable to the consumer.

The results of the organoleptic analysis showed significant differences in the attributes of mouth texture, color, and taste but not in the attributes of odor. Figure 1 shows the evaluations of the panel of tasters in the texture in the mouth, where the treatment “Wild + Quinoa flour + Sunflower oil “obtained 4.2 points, placing it with an excellent firm texture in the mouth. In the study by [28], no difference was observed in the attributes of texture and color. However, they observed a flavor difference with an average of 4.22 points. These authors state that fish flesh is inconsistent or firm enough to guarantee an adequate texture, so this attribute does not present an adequate significance according to their treatments. Other authors, such as [37], made sausages partially replacing beef with hake, reporting rejection by the panelists who did not like the soft texture, arguing that this characteristic is one of the major drawbacks when seafood pulp is added to the sausage-making process. Likewise, the results obtained in this research differ from those reported by [38], who observed lower acceptance in formulations containing a higher proportion of fish in sausages made with hake paste, attributing this to the strong fishy taste and smell undesired by tasters, which is much more intense as the percentage of fish used becomes higher. Similarly, [39] reported a higher acceptance of sausages with lower fish content in sausages made by partially replacing beef with tuna meat [40].

These results are in line with those described by [30] for Panga sausage, while the ratings described for tilapia sausages by [11] are much lower. Overall acceptability is 3.2 on a scale of 0–4, making the Old Blue (*A. rivulatus*) sausage more palatable to the consumer. 

The shelf life of this product, estimated by sensory analysis, was 30 days, verifying that it was within the parameters established microbiologically by [23].

## 4. Conclusions

In the data obtained for the content of saturated fatty acids present in sausages, highly significant differences could be observed concerning the rearing system (wild/farmed). Saturated and monosaturated fatty acids display higher concentrations in wild systems, but polyunsaturated fatty acid content was higher for the farmed species.

Highly significant differences were observed in all formulas, where the protein factor was one of the most important, obtaining the highest percentage with the addition of soybean flour with a value of 11.24%. The combination of factors “Wild + Quinoa flour + Sunflower oil” obtained the best sensory responses and nutritional characteristics of the lipid profile due to omega 3, 6, and 9 content and presented better emulsion consistency of the sausage. Microbiological analysis of sausage showed the absence of pathogenic microorganisms due to the adequate hygienic process. 

Sensory analysis established a shelf life of the sausage of 30 days, demonstrating added value through this processing. This allows its exploitation in areas further away from the fishing zone and, with it, a greater possibility of social development in the region, which will strategically allow us to achieve Sustainable Development Goals (SDGs 1, 2, 3, 5, 8, 12, 13, and 17), contributing to the development of a sustainable food production model and thus creating synergy towards new production lines with significant economic value.

Significant differences were observed between the rearing systems regarding sausage yield: the wild system was 62.4% and the farmed system was 81.7%. Therefore, it is justified that fish farm production is promoted as an alternative to the local generation of high-quality animal protein, which would benefit both human nutrition and the local economy of depressed areas. Moreover, this system is environmentally friendly and guarantees the survival of an autochthonous species that is sensitive to competition from other invasive species.

## Figures and Tables

**Figure 1 foods-13-01399-f001:**
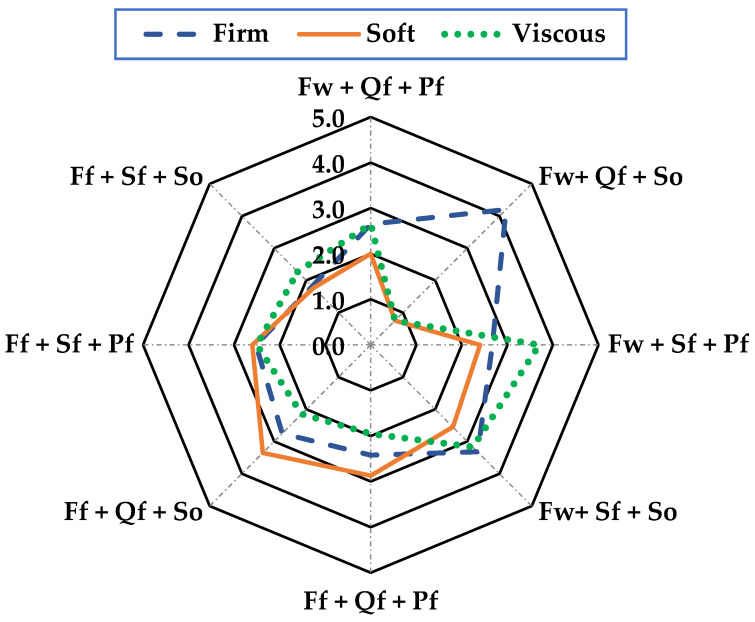
Texture in mouth evaluation of sausages of A. rivulatus. **Ff**: Fish farm, **Fw**: Fish wild; **Sf**: Soya flour, **Qf**: Quinoa flour; **Pf**: Pork fat, **So**: sunflower oil.

**Figure 2 foods-13-01399-f002:**
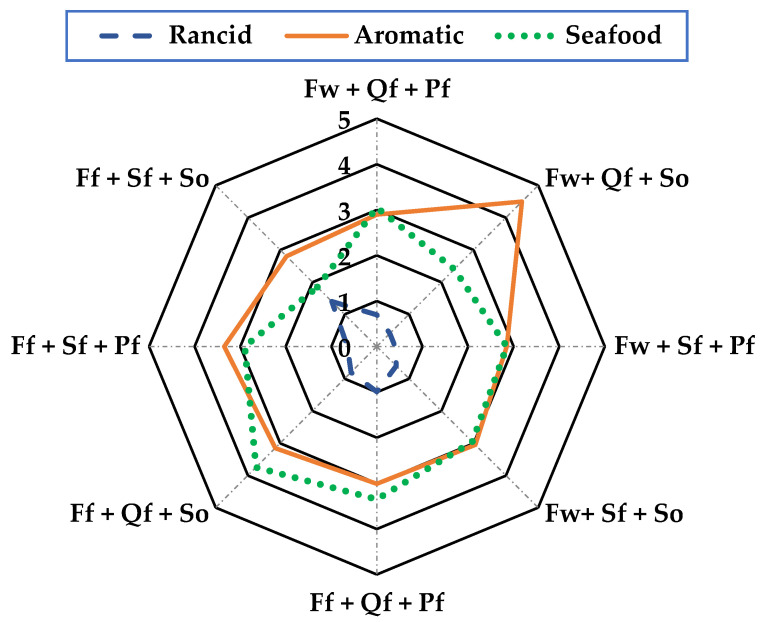
Smell in mouth evaluation of sausages of *A. rivulatus*. **Ff**: Fish farm, **Fw**: Fish wild; **Sf**: Soya flour, **Qf**: Quinoa flour; **Pf**: Pork fat, **So**: sunflower oil.

**Figure 3 foods-13-01399-f003:**
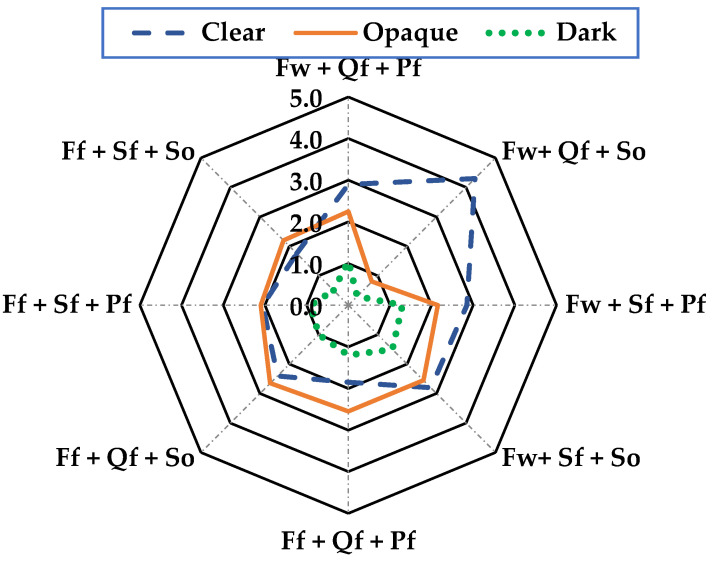
Color in mouth evaluation of sausages of A. rivulatus. **Ff**: Fish farm, **Fw**: Fish wild; **Sf**: Soya flour, **Qf**: Quinoa flour; **Pf**: Pork fat, **So**: sunflower oil.

**Figure 4 foods-13-01399-f004:**
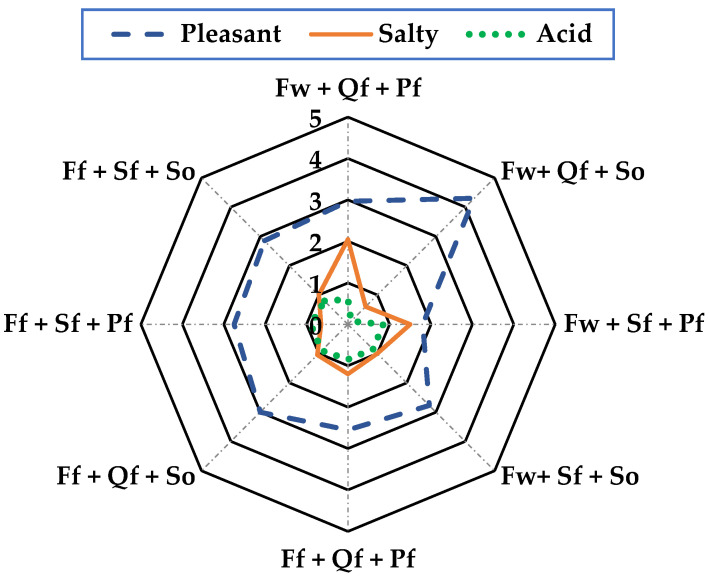
*Taste in mouth evaluation* of sausages *of A. rivulatus*. **Ff**: Fish farm, **Fw**: Fish wild; **Sf**: Soya flour, **Qf**: Quinoa flour; **Pf**: Pork fat, **So**: sunflower oil.

**Table 1 foods-13-01399-t001:** Formulation of fillet-based sausage Old Blue (*A. rivulatus*).

Raw Material	Weight (g)	Percentage (%)
Fish fillet (Wild/Farmed)	300.00	48.73
Lipids (sunflower oil/pork fat)	96.00	15.59
Protein (Quinoa flour/soy flour)	28.00	2.92
Salt Nitrite	1.80	0.30
Phosphate	1.20	0.19
Glutamate	1.00	0.16
Sodium chloride	6.00	0.97
Ascorbic acid	1.20	0.19
Starch	37.80	6.14
Liquid smoke	0.80	0.13
Seasonings	7.80	1.27

Seasonings: garlic, cumin, pepper, paprika, red onion powder.

**Table 2 foods-13-01399-t002:** Values of the immediate components of the sausage Old Blue (*A. rivulatus*) and significance according to the rearing system.

Rearing System	pH	Humidity %	Ash %	Protein (g/100)	Fat %
Wild	6.14 ^B^ ± 0.14	63.82 ^B^ ± 5.0	2.49 ^A^ ± 0.12	10.07 ^A^ ± 1.8	6.13 ^A^ ± 0.04
Farmed	5.84 ^A^ ± 0.20	62.77 ^A^ ± 8.5	2.48 ^A^ ± 0.04	12.55 ^B^ ± 2.0	6.09 ^A^ ± 0.09

^A^,^B^: statistically significant (using Tukey’s test *p* < 0.05).

**Table 3 foods-13-01399-t003:** Values (averages ± SD) of the immediate components of the sausage Old Blue (*A. rivulatus*) and significance according to the type of formula.

Formula	pH	Humidity %	Ash %	Protein (g/100)	Fat %
Wild + Quinoa flour+ Pork fat	6.12 ^E^ ± 0.11	66.06 ^G^ ± 4.95	2.49 ^C^ ± 0.39	10.06 ^B^ ± 0.30	7.58 ^F^ ± 1.02
Wild + Quinoa flour+ Sunflower oil	6.23 ^G^ ± 0.15	61.11 ^C^ ± 2.98	2.10 ^B^ ± 0.39	10.36 ^C^ ± 0.29	6.56 ^D^ ± 1.87
Wild + Soja flour + Pork fat	6.08 ^D^ ± 0.08	64.09 ^E^ ± 0.54	2.49 ^C^ ± 0.11	10.07 ^B^ ± 0.61	4.69 ^A^ ± 0.97
Wild + Soja flour+ Sunflower oil	6.16 ^F^ ± 0.15	63.55 ^D^ ± 2.97	2.50 ^C^ ± 0.98	8.60 ^A^ ± 1.47	5.66 ^C^ ± 2.11
Farmed + Quinoa flour+ Pork fat	6.01 ^C^ ± 0.19	66.52 ^H^ ± 8.40	1.52 ^A^ ± 1.45	11.23 ^D^ ± 1.16	7.77 ^G^ ± 1.04
Farmed + Quinoa flour+ Sunflower oil	5.82 ^AB^ ± 0.03	58.12 ^A^ ± 6.90	2.97 ^D^ ± 0.48	12.69 ^F^ ± 1.46	4.62 ^A^ ± 2.27
Farmed + Soja flour+ Pork fat	5.85 ^B^ ± 0.04	65.02 ^F^ ± 4.49	2.49 ^C^ ± 0.35	12.97 ^G^ ± 0.28	6.89 ^E^ ± 1.80
Farmed + Soja flour+ Sunflower oil	5.81 ^A^ ± 0.31	60.53 ^B^ ± 1.04	2.48 ^C^ ± 0.36	12.40 ^E^ ± 0.57	5.09 ^B^ ± 2.49

^A–G^: statistically significant using Tukey’s test (*p* < 0.05).

**Table 4 foods-13-01399-t004:** Saturated fatty acids in fish sausage Old Blue (*A. rivulatus*) and significance according to the rearing system.

		Rearing System	
		Wild	Farmed
Lauric acid	C12:0	0.31 ^A^	0.24 ^A^
Tridecanoic acid	C13:0	0.00 ^A^	0.26 ^B^
Myristic acid	C14:0	3.97 ^B^	2.16 ^A^
Pentanoic acid	C15:0	0.95 ^A^	1.25 ^A^
Palmitic acid	C16:0	19.04 ^B^	15.31 ^A^
Heptanoic acid	C17:0	1.53 ^A^	1.38 ^A^
Stearic acid	C18:0	4.74 ^A^	5.03 ^A^
Arachidic acid	C20:0	1.23 ^A^	1.09 ^A^
Heneicosane acid	C21:0	0.34 ^A^	0.18 ^A^
Behenic acid	C22:0	1.51 ^A^	2.23 ^B^
Total Saturated acids		33.66 ^B^	29.13 ^A^

^A^,^B^: statistically significant (*p* < 0.05).

## Data Availability

Data will be made available upon request.

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
