# Peer review of "Nutritional and Organoleptic Characteristics of Sausage Based on an Autochthonous Ecuatorian Fish, Old Blue (Andinoacara rivulatus)"

_foods, 2024, doi:10.3390/foods13091399_

Round 1
Reviewer 1 Report
Comments and Suggestions for Authors
Dear, considering the manuscript after revision my opinion is that the research seems to be interesting, provided by socio-economic impacts.
The title of the manuscript must be improved to reflect the innovative component of the manuscript. The introduction provides essential components of importance and relevance of the investigation but must be better structurally organized in logical order.
The aim of the study is well explained.
Nutritional, organoleptic, and microbiological characteristics must be explained. The Ecuadorian Technical Standards are the starting point of the used methodology, but details of the issued methodology must be provided.
Statistical analysis must be improved in detail followed by presented results.
The quality of the presentation satisfied the scientific criteria.
Conclusion justified obtained results.
Major revision.
Author Response
RESPUESTAS A LOS REVISORES
Agradecemos sinceramente a los revisores por sus comentarios constructivos que, en nuestra opinión, han mejorado la calidad y claridad del manuscrito revisado. Por favor, encuentre a continuación una respuesta detallada punto por punto a sus consultas.
Medina et al. en Características nutricionales y organolépticas de embutido a base de un pescado autóctono ecuatoriano, Old Blue (Andinoacara rivulatus) Manuscrito ID alimentos-2948442, donde este estudio tuvo como objetivo investigar las características sensoriales y nutricionales de un embutido elaborado a partir del filete del pescado nativo Old Blue. (Andinoacara rivulatus) para estimular la economía local. El análisis microbiológico del embutido con especies nativas mostró la ausencia de microorganismos patógenos con una vida útil comercial de 30 días, demostrando un valor agregado a través de este procesamiento, permitiendo su explotación en zonas más alejadas de los sitios de pesca y, por tanto, una mayor posibilidad. de desarrollo social en la zona.
Revisor 1
Respuesta al revisor 1 comentarios
Comentarios y sugerencias para autores
Estimado,
Después de revisar el manuscrito, creo que la investigación parece interesante, especialmente los impactos socioeconómicos.
- El título del manuscrito debe mejorarse para reflejar el componente innovador del manuscrito.
Respuesta 1.-
Se ha corregido en el texto según recomendación del revisor “ Características nutricionales y organolépticas de embutido a base de un pescado autóctono ecuatoriano, Old Blue ( Andinoacara rivulatus )”
- La introducción proporciona componentes esenciales de importancia y relevancia de la investigación, pero debe estar mejor organizada estructuralmente en un orden lógico. El objetivo del estudio está bien explicado.
Respuesta 2.-
Los autores agradecen al revisor sus comentarios positivos sobre la calidad del artículo y han mejorado la redacción para que sea más fácil de entender.
- Se deben explicar las características nutricionales, organolépticas y microbiológicas. Las Normas Técnicas Ecuatorianas son el punto de partida de la metodología utilizada, pero se deben brindar detalles de la metodología emitida.
Respuesta 3.-
El texto ha sido corregido según recomendación del revisor.
- El análisis estadístico debe mejorarse en detalle seguido de los resultados presentados.
Respuesta 4.-
El texto ha sido corregido según recomendación del revisor.
- La calidad de la presentación satisfizo los criterios científicos y la conclusión justificó los resultados obtenidos.
Respuesta 5.-
Los autores agradecen al revisor por sus comentarios positivos sobre la calidad del artículo.

Reviewer 2 Report
Comments and Suggestions for Authors
The manuscript entitled “Nutritional and organoleptic characteristics of Old Blue sausage (Andinoacara rivulatus)” investigate the sensory and nutritional characteristics of a sausage made from the fillet of the native fish Old Blue (Andinoacara rivulatus), I think it’s a quiet interesting topic, but some issue still should be solved.
1. Firstly the author needs to check the entire text because there are many strange paragraphs.
2. The introduction part does not clearly introduce the purpose and significance of this study, and needs to be revised.
3. The detail information about materials should be given
4. The results should be show as AVR±SD
5. What is wild and farmed? it should be mentioned in both introduction and materials
6. Conclusion should be rewritten, it need be more concise.
Author Response
Revisor 2
Respuesta al revisor 2 comentarios
Comentarios y sugerencias para autores.
Comentarios específicos
El manuscrito titulado “Características nutricionales y organolépticas de la salchicha Old Blue ( Andinoacara rivulatus )” investiga las características sensoriales y nutricionales de una salchicha elaborada a partir del filete del pescado nativo Old Blue ( Andinoacara rivulatus ), creo que es un tema bastante interesante, pero algún problema aún debe resolverse.
- En primer lugar, el autor debe revisar el texto completo porque hay muchos párrafos extraños.
Respuesta 6.-
Se ha corregido en el texto según recomendación del revisor.
- La parte de introducción no presenta claramente el propósito y la importancia de este estudio y necesita ser revisada.
Respuesta 7.-
Los autores agradecen al revisor por sus comentarios positivos sobre la calidad del artículo. Según recomendación del revisor, el texto ha sido modificado para facilitar la comprensión y lectura del artículo.
.
- Se debe proporcionar información detallada sobre los materiales.
Respuesta 8.-
El texto ha sido corregido según recomendación del revisor.
- Los resultados deben mostrarse como AVR±SD
Respuesta 9.-
Se ha corregido en la Tabla según recomendación del revisor.
- ¿Qué es silvestre y cultivado? debe mencionarse tanto en la introducción como en los materiales.
Respuesta 10.-
El texto ha sido corregido según recomendación del revisor.
- La acuicultura es el cultivo de organismos acuáticos, ha sido la actividad agroindustrial con mayor tasa de crecimiento a nivel mundial en las últimas cuatro décadas (https://doi.org/10.1100/2012/389623). Esta actividad se puede realizar en instalaciones artificiales o no, aunque lo más destacable es que se controlan todas las etapas del desarrollo en condiciones óptimas (se seleccionan los recursos reproductivos, se recogen los huevos y se tabulan los alimentos y se seleccionan otros factores ambientales y productivos).
-El término silvestre comprende aquellas poblaciones silvestres que habitan los ríos de forma natural, sin intervención del hombre para su producción, permitiendo una considerable variabilidad de las especies silvestres.
“Las muestras se tomarían de dos sistemas de cría: Acuicultura que es el cultivo de organismos acuáticos, cuya actividad puede realizarse en instalaciones artificiales o no, aunque lo más destacable es que todas las etapas de desarrollo están controladas en condiciones óptimas; y el sistema de pesca artesanal está formado por aquellas poblaciones silvestres que habitan los ríos de forma natural, sin intervención del hombre para su producción, permitiendo una variabilidad considerable de las especies silvestres”.
- La conclusión debería reescribirse, debe ser más concisa.
Respuesta 11.-
Se ha corregido en el texto según recomendación del revisor.

Reviewer 3 Report
Comments and Suggestions for Authors
The manuscript entitled “Nutritional and organoleptic characteristics of Old Blue Sausage (Andinoacara rivulatus)” aimed to characterize the nutritional and organoleptic properties of fish sausage made from autochthonous Ecuadorian fish. The presented work is well organized; the research was conducted systematically but has several major deficiencies that are listed below.
Line 77: Experimental design: The authors must clarify the experimental design. It is not clear how many individual fish were collected on each location, how often the samples were collected, how many fish fillets were pooled for the production of one sausage.
Line 139–142: The sausage's final weight yield is 76,33% and this is not in accordance with the conclusion (Line 292-293). The authors must change results section or change the conclusion
The overall statistical interpretation is poor, while it is not clear how statistically significant differences are interpreted—between columns or rows?
Author Response
Reviewer 3
Response to Reviewer 3 Comments
Comments and Suggestions for Authors
Specific comments
The manuscript entitled “Nutritional and organoleptic characteristics of Old Blue Sausage (Andinoacara rivulatus)” aimed to characterize the nutritional and organoleptic properties of fish sausage made from autochthonous Ecuadorian fish. The presented work is well organized; the research was conducted systematically but has several major deficiencies that are listed below.
- Line 77: Experimental design: The authors must clarify the experimental design. It is not clear how many individual fish were collected on each location, how often the samples were collected, how many fish fillets were pooled for the production of one sausage.
Response 12.-
It has been corrected in the text according to the reviewer's recommendation. Line 79-92
“2.1. Experimental design
The wild samples were collected in artisanal fishing areas used by local people, who were the collectors and collaborators in this research. Artisanal fishing obtained wild specimens of old blues from rivers of the Baba (Buena Fe canton) and the "La Estrella"(Mocache canton) sector, and the farmed fish were situated in Puerto Bajaña (Mocache canton) and the D'Veritas (Mocache canton) sector, Puerto Bajaña, all of them in Los Rios province.
An equal number of fish were collected monthly from each rearing system, for a total of 72 fish during the winter period from October to December.
After being eviscerated, washed, and cut into pieces, samples of fish fillets were stored in airtight bags at -2°C.
A multifactorial design (A*B*C) was applied, where each factor consists of two levels: Factor A = Rearing system (Wild / Fish farm), Factor B = Type of protein (Quinoa meal / Soybean meal) and Factor C = Type of lipid (Pig fat / Sunflower oil), obtaining a total of eight formula. Each treatment was replicated three times.“
- Line 139–142: The sausage's final weight yield is 76,33% and this is not in accordance with the conclusion (Line 292-293). The authors must change results section or change the conclusion
Response 13.-
It has been corrected in the text according to the reviewer's recommendation.
- The overall statistical interpretation is poor, while it is not clear how statistically significant differences are interpreted—between columns or rows?
Response 14.-
The difference between rows is detailed in the titles of the tables: Table 2. Values of the immediate components of the sausage Old Blue (A. rivulatus) and significance according to the rearing system. Table 3. - Values of the immediate components of the sausage Old Blue (A. rivulatus) and significance according to the type of formula.

Round 2
Reviewer 1 Report
Comments and Suggestions for Authors
Dear, after carefully considering of revised manuscript, I can conclude that some improvements have been made, but the Introduction and statistical methodology were not sufficiently improved according to suggestions.
Therefore, I suggest to the editor to decisions regarding that.
Author Response
The reviewer was too unspecific in this matter, so we don’t really know what he /she considers logical order.
The Introduction has the following structure of paragraph :
It discusses the increasing consumption of fisheries and aquaculture products and the importance of sustainable aquaculture development, then highlights Ecuador's position as a major fishing and exporting country.
-Addresses the issue of overconsumption, which is endangering native species like the Old Blue and requiring interventions to prevent extinction, and describes afterward the current situation of fishing in Ecuador, especially concerning artisanal fishing and the breeding of native fish.
-Mentions the significance of previous studies on fish production and feed composition in aquaculture, as well as the existing research gaps regarding the Old Blue and the use of local products. This links with the next paragraph, which emphasizes the nutritional value of fish in the human diet and its role in preventing cardiovascular diseases.
-Explores the possibility of using fish in more acceptable and accessible food products, such as sausages. the next paragraph summarizes previous studies on fish-based sausages, mentioning the species investigated so far and highlighting the lack of studies on the Old Blue in this context.
Finally, describes the proposal to use Andinoacara rivulatus fillets to make Frankfurt-type sausages. Details the factors considered in sausage production, such as breeding systems, vegetable proteins, and lipids used.
species.
This structure provides a logical and coherent order for the text, guiding readers through the context, objectives, methodology, results, and implications of the research on Old Blue-based sausages in Ecuador.
The statistical analysis was incorporated in the results text and tables. The reviewer was too unspecific in this matter so we hope this is enough.
Reviewer 2 Report
Comments and Suggestions for Authors
No comment
Reviewer 3 Report
Comments and Suggestions for Authors
The authors made all required changes so manuscript can be accepted in present form.
Author Response
Authors are very grateful of the comments of reviewer: The authors made all required changes so manuscript can be accepted in present form.